# Structural basis for membrane recruitment of ATG16L1 by WIPI2 in autophagy

Lisa M Strong[1,2,3], Chunmei Chang[1,2,3], Julia F Riley[3,4], C Alexander Boecker[3,4], Thomas G Flower[1,2†], Cosmo Z Buffalo[1,2], Xuefeng Ren[1,2], Andrea KH Stavoe[5], Erika LF Holzbaur[3,4], James H Hurley[1,2,3]*

[1]Department of Molecular and Cell Biology, University of California, Berkeley, Berkeley, United States; [2]California Institute for Quantitative Biosciences, University of California, Berkeley, Berkeley, United States; [3]Aligning Science Across Parkinson's (ASAP) Collaborative Research Network, Chevy Chase, United States; [4]Department of Physiology, University of Pennsylvania Perelman School of Medicine, Philadelphia, United States; [5]Department of Neurobiology and Anatomy, The University of Texas Health Science Center at Houston McGovern Medical School, Houston, United States

**ABSTRACT** Autophagy is a cellular process that degrades cytoplasmic cargo by engulfing it in a double-membrane vesicle, known as the autophagosome, and delivering it to the lysosome. The ATG12–5–16 L1 complex is responsible for conjugating members of the ubiquitin-like ATG8 protein family to phosphatidylethanolamine in the growing autophagosomal membrane, known as the phagophore. ATG12–5–16 L1 is recruited to the phagophore by a subset of the phosphatidy-linositol 3-phosphate-binding seven-bladedß -propeller WIPI proteins. We determined the crystal structure of WIPI2d in complex with the WIPI2 interacting region (W2IR) of ATG16L1 comprising residues 207–230 at 1.85 Å resolution. The structure shows that the ATG16L1 W2IR adopts an alpha helical conformation and binds in an electropositive and hydrophobic groove between WIPI2 ß-propeller blades 2 and 3. Mutation of residues at the interface reduces or blocks the recruitment of ATG12–5–16 L1 and the conjugation of the ATG8 protein LC3B to synthetic membranes. Interface mutants show a decrease in starvation-induced autophagy. Comparisons across the four human WIPIs suggest that WIPI1 and 2 belong to a W2IR-binding subclass responsible for localizing ATG12–5–16 L1 and driving ATG8 lipidation, whilst WIPI3 and 4 belong to a second W34IR-binding subclass responsible for localizing ATG2, and so directing lipid supply to the nascent phagophore. The structure provides a framework for understanding the regulatory node connecting two central events in autophagy initiation, the action of the autophagic PI 3-kinase complex on the one hand and ATG8 lipidation on the other.

*For correspondence: jimhurley@berkeley.edu

Present address: †Galapagos, Romainville, France

## Introduction

Macroautophagy (hereafter autophagy) maintains cellular homeostasis by sequestering unneeded or harmful cytoplasmic material in double-membrane vesicles known as autophagosomes (*Morishita and Mizushima, 2019*). Mature autophagosomes fuse with lysosomes, leading to degradation of engulfed contents. Starvation-induced autophagy is thought to target bulk cytosol, while various forms of selective autophagy target damaged mitochondria and other organelles, invading bacteria, protein aggregates, and many other intracellular materials (*Anding and Baehrecke, 2017*; *Gomes and Dikic, 2014*). Defects in autophagy are associated with increased vulnerability to pathogens, aging, and neurodegenerative diseases (*Levine and Kroemer, 2019*). Defects in the autophagy of

mitochondria ('mitophagy') downstream of Parkin and PINK1 are associated with hereditary early onset Parkinson's disease (*Pickrell and Youle, 2015*; *Stavoe and Holzbaur, 2019*).

The many varieties of bulk and selective autophagy all rely on a handful of shared core components, which include the class III phosphatidylinositol 3-kinase complex I (PI3KC3-C1); the ubiquitin-like ATG8 family (LC3A-C, GABARAP, and GABARAPL1-2 in mammals); the proteins ATG7, ATG3, and ATG12–5-16 L1 responsible for conjugating ATG8s to phosphatidylethanolamine (PE); and the WD-repeat protein interacting with phosphoinositide (WIPI family) (*Chang et al., 2021a*; *Mizushima et al., 2011*). PI3KC3-C1 is targeted to sites of autophagy initiation by its ATG14 subunit, where it phosphorylates phosphatidylinositol (PI) at the third position in the inositol ring to generate PI(3)P (*Itakura et al., 2008*; *Obara et al., 2006*; *Sun et al., 2008*). ATG8 proteins are attached to the membrane lipid phosphatidylethanolamine (PE) in a process that is closely analogous to the conjugation of ubiquitin to its target proteins (*Ichimura et al., 2000*). In brief, ATG4 cleaves ATG8 to expose the C-terminal glycine, the ubiquitin E1-like ATG7 then activates ATG8 for transfer to the ubiquitin E2-like ATG3, and the ATG12–5-16 L1 complex scaffolds the ATG8 transfer from ATG3 to the headgroup of PE (*Klionsky and Schulman, 2014*). The function of ATG12–5-16L1 is analogous to that of ubiquitin E3 ligases, and we therefore refer to this complex here as 'E3'. This process is often referred to as LC3 lipidation, after LC3, the founding member of the ATG8 family in mammals (*Kabeya et al., 2000*). In mammals, ATG8 conjugation to membranes is important for multiple steps in autophagy and is particularly critical for autophagosome-lysosome fusion (*Nguyen et al., 2016*; *Tsuboyama et al., 2016*).

The two critical steps in autophagy initiation, PI 3-phosphorylation and LC3 lipidation, are connected to one another via a direct interaction between a subset of the PI(3)P-binding WIPI proteins and ATG16L1 (*Dooley et al., 2014*). The human WIPI1-4 proteins comprise a subset of the seven bladed β-propeller protein binding to phosphoinositides (PROPPINs) (*Dove et al., 2004*). PROPPINs bind to PI(3)P and PI(3,5)$P_2$ headgroups through a conserved FRRG motif (*Dove et al., 2004*; *Gaugel et al., 2012*) and bind tightly, but reversibly, to membranes using a hydrophobic loop in blade six that inserts into the membrane (*Baskaran et al., 2012*; *Krick et al., 2012*; *Watanabe et al., 2012*). WIPI2 is expressed as six known isoforms, which appear to have overlapping functions (*Proikas-Cezanne et al., 2015*). WIPI2b in particular has been shown to have a central role in bulk and selective autophagy initiation in cells (*Dooley et al., 2014*; *Polson et al., 2010*), and WIPI2d potently activated LC3 lipidation in an in vitro giant unilamellar vesicle (GUV) reconstituted system (*Fracchiolla et al., 2020*).

Despite the centrality of the WIPI2:ATG16L1 interaction to mammalian autophagy initiation, only a predictive model (*Dooley et al., 2014*), but no experimentally determined structure has been available. Here, we report the crystal structure of WIPI2d:ATG16L1 (207–230) complex at a 1.85 Å resolution. WIPI2d point mutations in the interface disrupted ATG16L1 binding, reduced the ability of WIPI2 to recruit ATG12–5-16 L1 and promote LC3 lipidation on GUVs, and reduced starvation-induced autophagy in cells.

## Results

### Structure determination of WIPI2d:ATG16L1-W2IR

In order to generate a crystallizable form of WIPI2d, the flexible hydrophobic loop in blade six and the putatively disordered C-terminal region were deleted (*Figure 1A*). The deletion construct removes the only regions whose sequence diverges between WIPI2b and WIPI2d; thus, the construct represents a WIPI2b/d consensus. A peptide corresponding to the WIPI2-interacting region ('W2IR') comprising residues 207–230 of ATG16L1 (*Dooley et al., 2014*) was synthesized. The crystal structure of the WIPI2d:ATG16L1 complex was determined at 1.85 Å (*Figure 1B,C*) by molecular replacement using the structure of *Kluveromyces lactis* Hsv2 (*Baskaran et al., 2012*) (PDB: 4EXV) as a search model. ATG16L1 was modeled de novo into the density (*Figure 1D*). The asymmetric unit contains two copies of the WIPI2d:ATG16L1 W2IR complex. One WIPI2d monomer is bound to one ATG16L1 peptide, and the two copies align with a Cα root-mean-square deviation (RMSD) of 0.3 Å. Statistics of crystallographic data collection and structure refinement are provided in *Supplementary file 1*. As expected on the basis of the Hsv2 (*Baskaran et al., 2012*; *Krick et al., 2012*; *Watanabe et al., 2012*) and WIPI3 (*Ren et al., 2020*) structures, WIPI2d folds into a seven blade ß -propeller, with each blade containing four anti-parallel ß-strands. The propeller is ~50 Å wide and ~30 Å tall (*Figure 1B,C*). The FRRG motif that enables WIPI2d binding to phosphoinositides is distal to the ATG16L1 binding site.

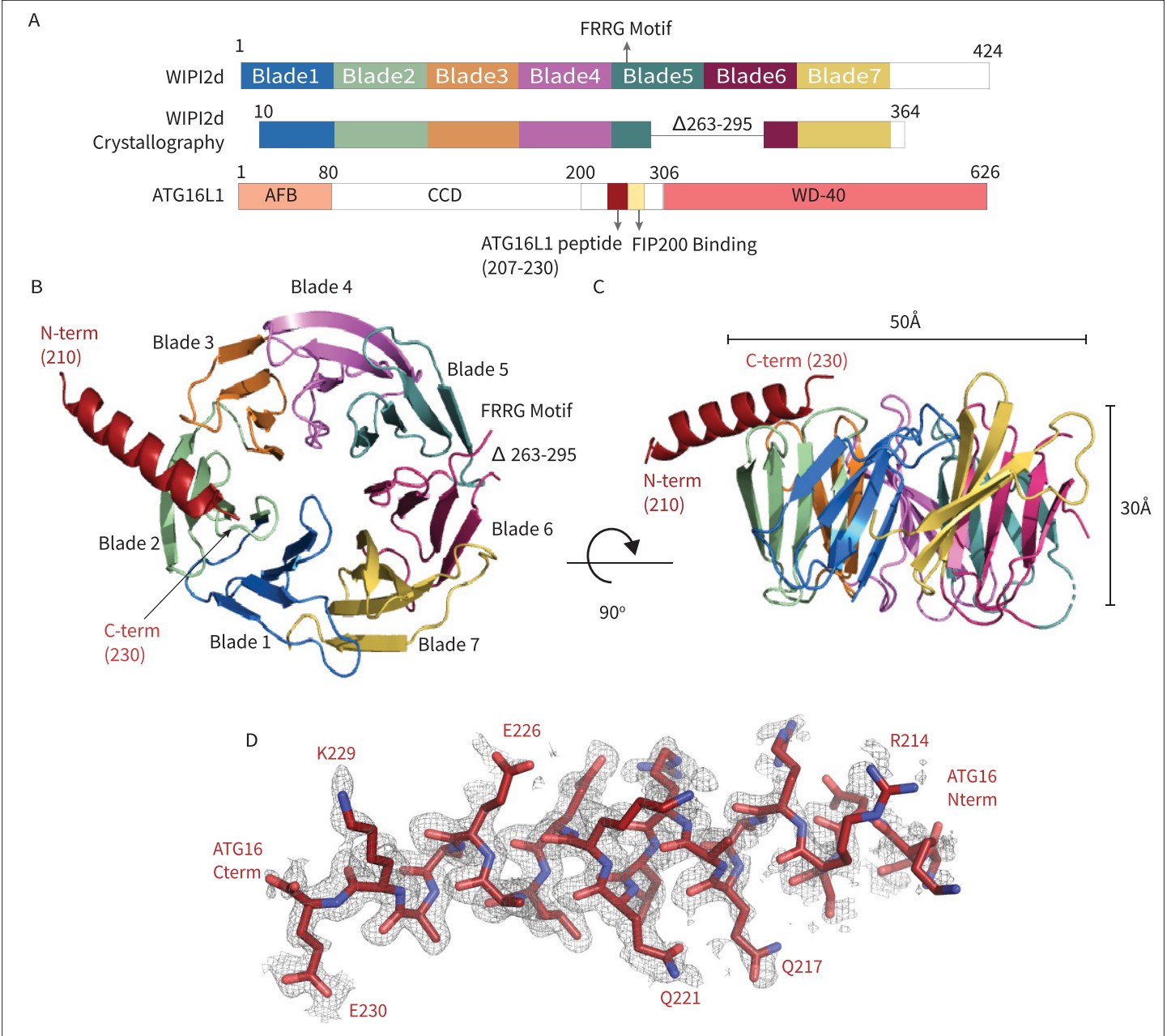

**Figure 1.** WIPI2d:ATG16L1 W2IR structure. Structure of WIPI2d bound to ATG16L1 W2IR. (**A**) Annotated WIPI2d and ATG16L1 domain schematics. WIPI2d construct for crystallography is shown and W2IR from ATG16L1. (**B, C**) The ribbon diagram of the WIPI2d complex with ATG16L1 W2IR from the (**B**) bottom and (**C**) side views. Each blade is colored in accordance with (**A**). (**D**) Composite omit map of ATG16L1 W2IR. Modeled ATG16L1 is shown as red carton and the composite omit 2mFo-DFc map contoured at 1σ is shown in gray.

## Analysis of WIPI2d W2IR: ATG16L1 interface

The ATG16L1 W2IR nestles between blades 2 and 3 of WIPI2d, burying ~550 Å² of solvent-accessible surface area. Blades 2 and 3 are identical in all six WIPI2 isoforms; thus, we expect that conclusions concerning the ATG16L1 binding mode drawn here will pertain to all WIPI2 isoforms. The WIPI2d binding site for the ATG16L1 W2IR consists of a single deep groove with a mixed electropositive and hydrophobic character (*Figure 2A,C*). Hydrophobic side chains of Leu 64, Phe 65, Leu 69, Val 83, Ile 92, Cys 93, Ile 124, and Met 127 on WIPI2d contribute to the hydrophobic surface of the groove. The surfaces of Leu 220 and Leu 224 of the ATG16L1 W2IR are buried in this interface (*Figure 2C,D*). The side chains of WIPI2d His 85, Lys 88, Arg 108, and Lys 128 contribute to the electropositive character

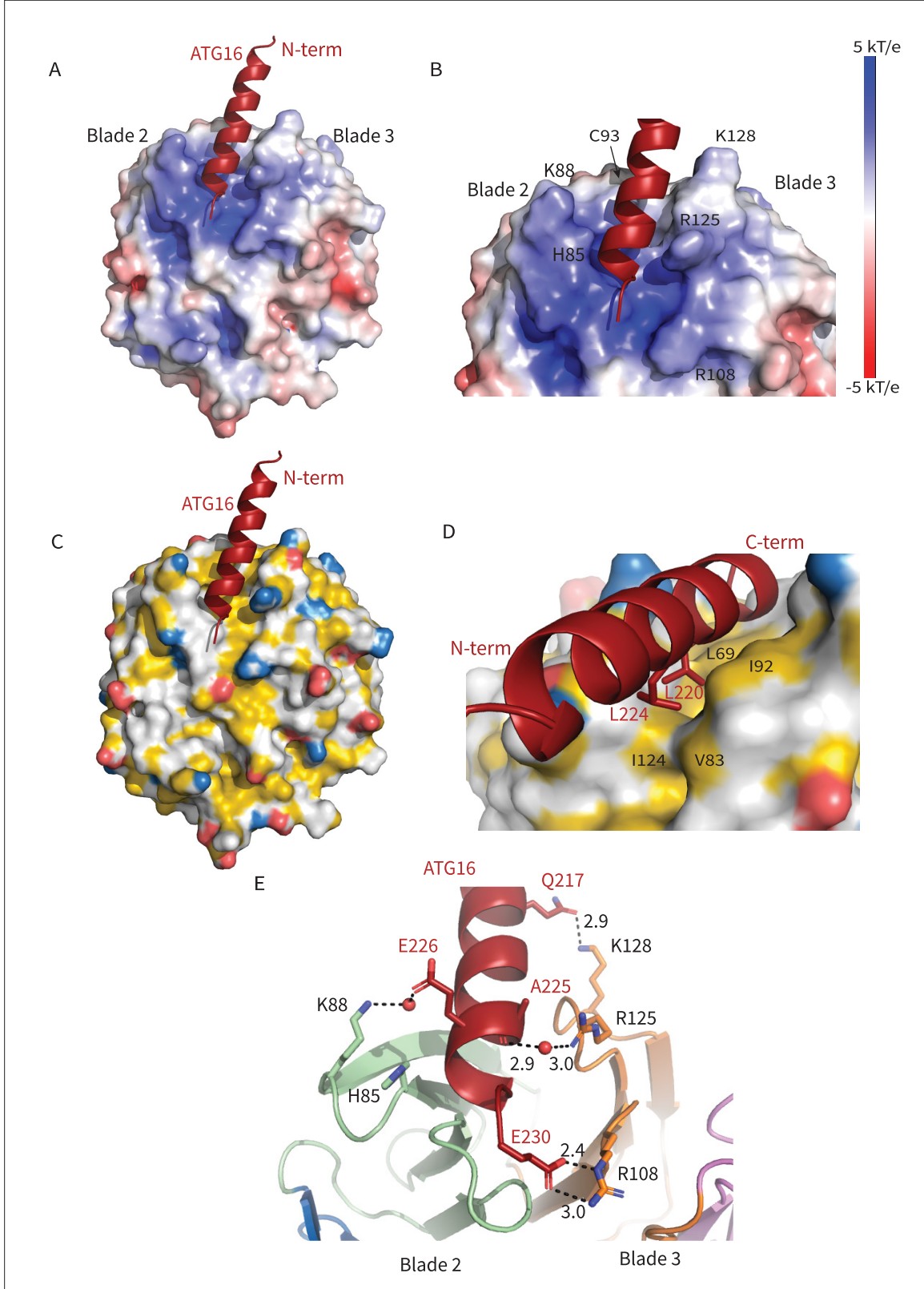

**Figure 2.** Interactions at the interface. Analysis of the Interface. (**A**) Overall electrostatic surface and (**B**) closer view of electrostatic surface with ATG16 W2IR shown as a cartoon and key residues labeled. (**C**) Overall hydrophobic surface of WIPI2d and (**D**) closer view of the hydrophobic interface with key residues labeled where yellow represents hydrophobic regions. (**E**) A cartoon and stick representation of hydrogen bonds between ATG16 and WIPI2d shown as black dotted lines with distances noted and key residues shown as sticks.

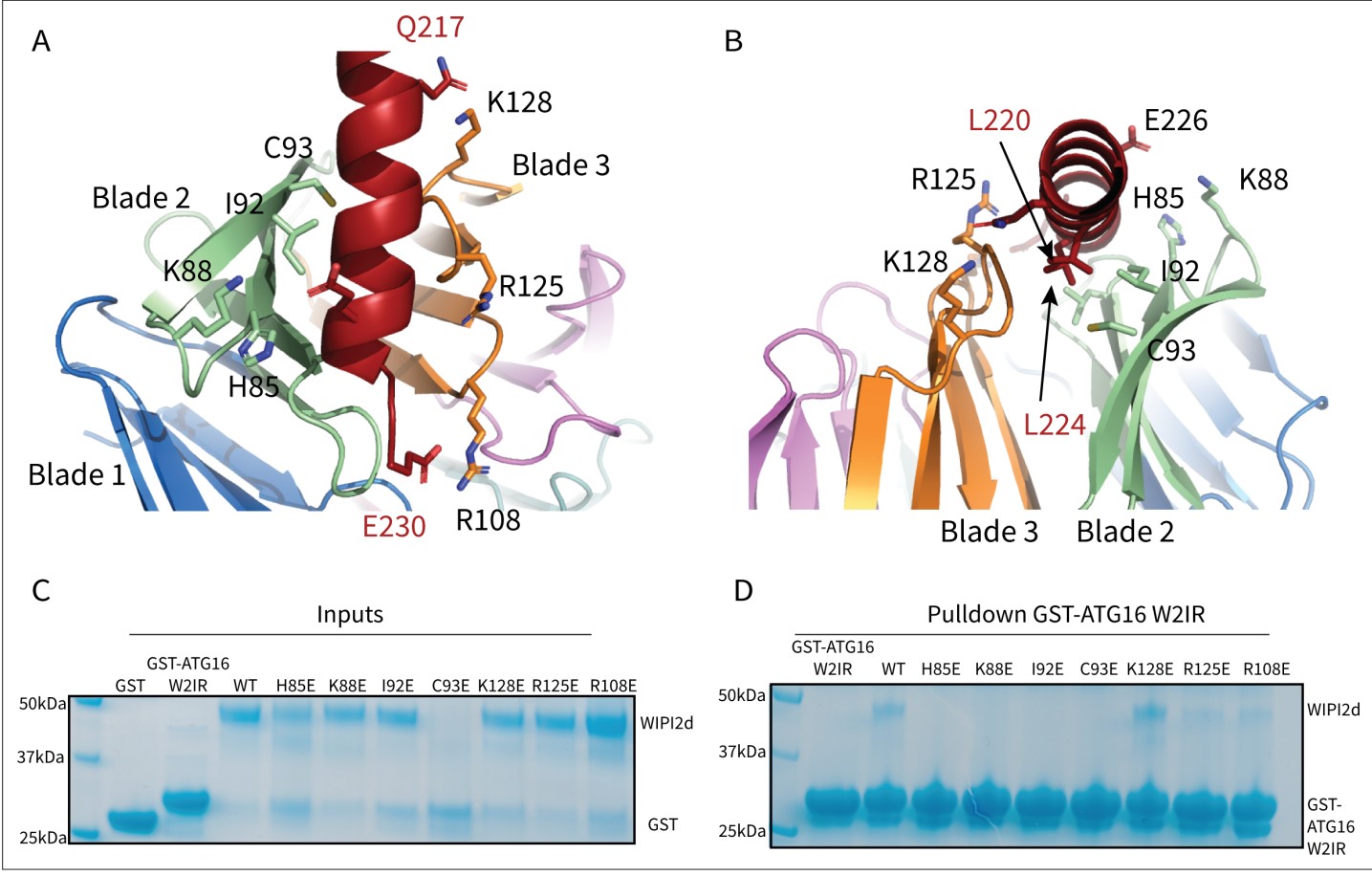

**Figure 3.** WIPI2d Interfacial mutants decrease ATG16L1 binding. Key interacting residues shown as sticks in cartoon representation of WIPI2d:ATG16L1 interface shown from (**A**) the WIPI2d face or (**B**) down the ATG16L1 helix. (**C**) Inputs for the (**D**) Pull-down assays of mutant WIPI2d constructs and wild type with GST-ATG16L1 W2IR. GSH resin was used to pull-down GST-ATG16L1 W2IR from purified protein mixture. The pull-down results were performed in triplicates and visualized by SDS–PAGE and Coomassie blue staining.

The online version of this article includes the following figure supplement(s) for figure 3:

**Source data 1.** Uncropped SDS-PAGE gels for *Figure 3*.

of the groove. The acidic side chains of Glu 226 and Glu 230 of ATG16L1 interact with the electropositive patch on WIPI2 (*Figure 2E*). The presence of WIPI2d Arg 108 and Arg 125, and ATG16L1 Glu 230 in the binding site was correctly predicted by the modeling efforts of Tooze and colleagues (*Dooley et al., 2014*). The nature of their interactions can now be defined on the basis of the crystal structure of the complex. Gln 217 of ATG16L1 forms a hydrogen bond with Lys 128 of WIPI2d at the N-terminus of the W2IR and WIPI2d, respectively. The C-terminus of the ATG16L1 W2IR, Glu 230 forms a salt bridge with Arg 108 of ATG16L1. Arg125 makes a water-mediated bridge to the W2IR peptide backbone in one of the two complexes in the asymmetric unit. Ser 66, Ser 67, and Ser 68 contribute additional polar interactions. The backbone of ATG16L1 near Ala 227 and Ala 228 forms a hydrogen bond with the backbone of WIPI2d between residues Ser 68 and Leu 69. This backbone binding favorably buries residues Leu 64, Phe 65, and Ser 67 within WIPI2d.

## Roles of WIPI2 interfacial residues

To evaluate the role of specific residues at the interface, we introduced single site mutations into WIPI2d to disrupt binding. H85E, K88E, and C93E were designed to perturb the electropositive WIPI2d surface on blade 2 (*Figures 2B and 3A,B*). L69E and I92E were designed to disrupt the hydrophobic groove for hydrophobic packing of ATG16L1 (*Figures 2 and 3Figures 2D and 3A,B*). K128E and R108E were chosen to abolish the interactions with Gln 217 and Glu 230 in ATG16L1, respectively (*Figures 2E and 3A,B*). R125E was designed to disrupt the bridging interaction to Lys88 (*Dooley*

*et al., 2014*). Both R108E and R125E were previously shown to reduce binding within the cellular context; thus, these two mutants also served to confirm that our in vitro binding experiments support the findings of previously reported immunoprecipitations (*Dooley et al., 2014*). To investigate the effects of these mutants on complex formation, we purified the mutant proteins and performed a coprecipitation assay using immobilized GST-ATG16L1 W2IR (*Figure 3C,D*). It was observed that L69E and C93E were prone to aggregation and were therefore not characterized further. All other mutants expressed at near identical levels as wild type, were purified at equivalent yields, and so presumed not to have grossly perturbed structures and stabilities. H85E, K88E, and I92E completely abolished binding to ATG16L1, while R108E and R125E retained weak binding to ATG16L1 (*Figure 3D*). Interestingly, K128E is coprecipitated at similar levels to WT WIPI2d (*Figure 3D*). Lys128 is positioned within a flexible loop (*Figure 3A*) near the location of three disordered Arg residues in the N-terminal part of the ATG16L1 W2IR preceding Gln 217. The resulting charge repulsion might offset the contribution of the W2IR Gln 217 hydrogen bond. The presence of these apparent negative interactions suggests that the association of the wild-type complex has evolved to a moderate affinity to facilitate the dissolution of the complex during the course of autophagosome maturation.

## The WIPI2d:ATG16L1 W2IR interface is required for LC3 lipidation in vitro

We next assessed the ability of WIPI2d mutants to activate E3 membrane recruitment and LC3 lipidation in a microscopy-based GUV assay (*Chang et al., 2021b*; *Fracchiolla et al., 2020*). In the presence of WIPI2d WT and the LC3 conjugation machinery (ATG7, ATG3, the ATG12–5-16 L, and a mCherry-LC3B construct corresponding to the ATG4-processed form) (*Figure 4A*), PI3KC3-C1 robustly triggered membrane recruitment of the E3-GFP complex and activated mCherry-LC3B lipidation (*Figure 4B,C*). Consistent with expectation, mutation of the previously characterized ATG16L1 binding sites R108E and R125E significantly reduced E3 membrane binding and LC3 lipidation (*Figure 4B,C*). The mutants H85E and I92E almost completely abolished E3 membrane binding and LC3 lipidation (*Figure 4B,C*). The K88E mutant also had an obvious defect in E3 recruitment and LC3 lipidation (*Figure 4B,C*). All of these observations are consistent with the loss of binding noted in the GST pull-down experiments. Consistent with the positive pull-down result, the K128E mutant fully retained the ability to recruit E3 to GUV membrane and activate subsequent LC3 lipidation (*Figure 4*). These data confirm that the ATG16L1 binding interface on WIPI2d is responsible for the E3 recruitment and activation that promotes LC3 membrane conjugation.

## Mutations that disrupt the WIPI2:ATG16L1 W2IR interface impair starvation-induced autophagy

These structural observations and in vitro reconstitutions cumulatively suggest that mutations disrupting the interface between WIPI2 and ATG16 might be expected to disrupt autophagosome formation. To test this hypothesis, we engineered Halo-tagged WIPI2B constructs containing H85E, K88E, and I92E mutations, as well as a construct containing all three of these mutations. In parallel, we expressed a Halo-tagged WIPI2B construct containing an R108E mutation, previously shown to disrupt the WIPI2B/ATG16 interaction (*Dooley et al., 2014*). Engineered WIPI2 constructs were expressed in WIPI2 KO HeLa cells generated by CRISPR/Cas9 gene editing and verified by Sanger sequencing and western blot (*Fischer et al., 2020*). The parent HeLa cell line was tested in parallel, transfected with a vector encoding the Halo-tag only. Autophagy was induced by incubating cells for 2 hr in 1 × EBSS (starvation media). All EBSS contained 100 nM of bafilomycin A (BafA) to block autophagosome/lysosome fusion.

Autophagosome number was scored to assess the impact of WIPI2 mutations on starvation-induced autophagy (*Figure 5A,B*). Compared to WIPI2B KO cells transfected with Halo-tagged WT WIPI2, autophagosome number was significantly lower in KO cells expressing the H85E ($p<0.0001$), I92E ($p<0.0001$), and triple mutant (H85E/K88E/I92E) ($p<0.0001$) constructs (n = 4). Each of these mutations caused a reduction in autophagosome formation comparable to that of the previously characterized R108E mutation; no statistically significant difference in autophagosome formation existed between the R108E construct and these mutations. Moreover, introducing WIPI2 containing these mutations lowered autophagosome number per cell significantly compared to WIPI2 KO cells (H85E, $p<0.01$; I92E, $p<0.01$; R108E, $p<0.01$; H85E/K88E/I92E, $p<0.01$), indicating these mutations

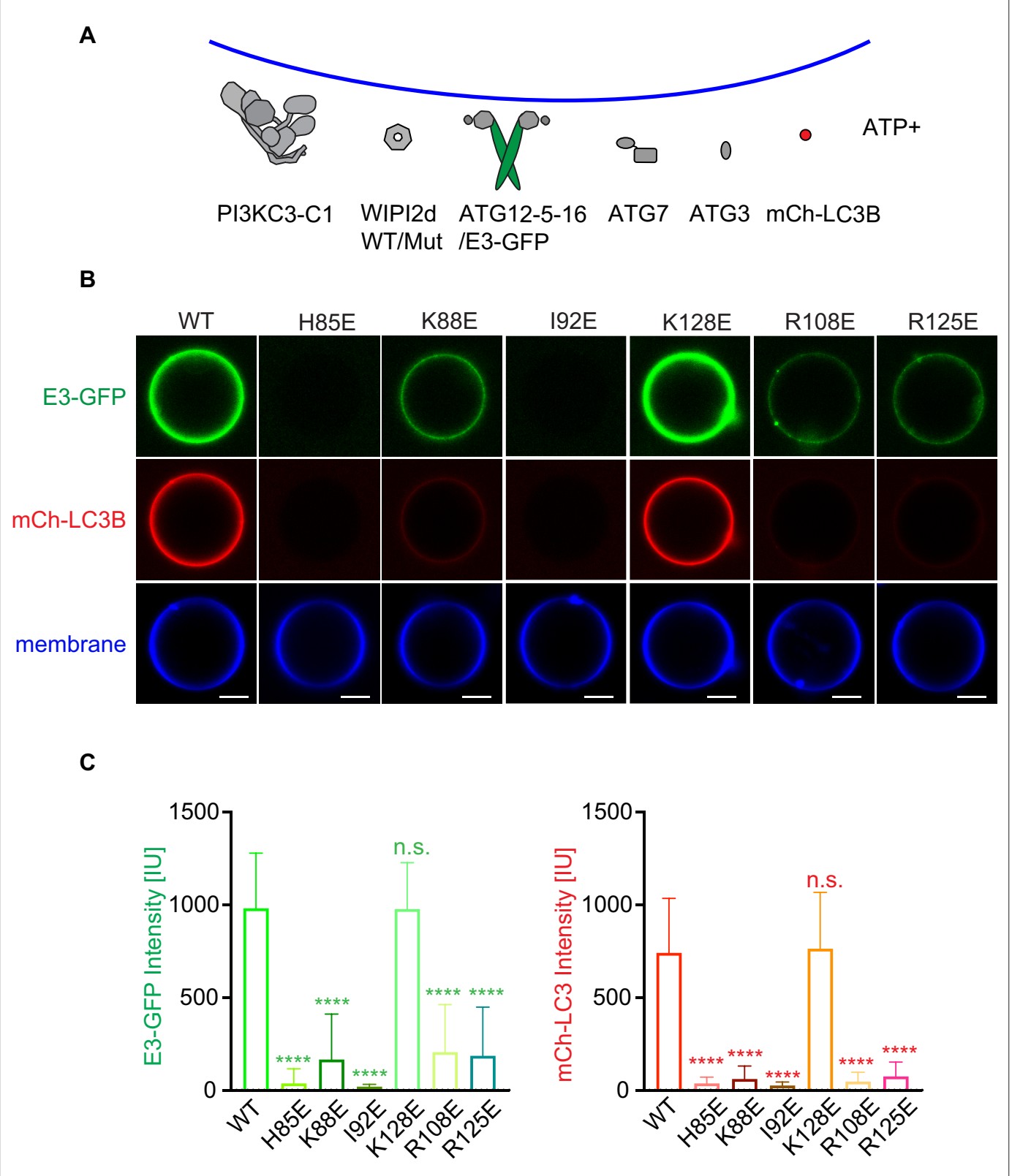

**Figure 4.** WIPI2d mutants disrupt E3 recruitment and LC3 lipidation on GUVs. (**A**) The schematic drawing illustrates the reaction setting. Colors indicate fluorescent protein-fused components. Components in gray are not labeled but are present in the reaction. (**B**) Representative confocal images of GUVs showing E3 membrane binding and LC3B lipidation. PI3KC3-C1, WIPI2d WT or mutant, E3-GFP, ATG7, ATG3, mCherry-LC3B, and ATP/Mn$^{2+}$ were incubated with GUVs (64.8 % DOPC: 20 % DOPE: 5 % DOPS: 10 % POPI: 0.2 % Atto647 DOPE) at room temperature. Images taken at 30 min were

*Figure 4 continued on next page*

*Figure 4 continued*

shown. Scale bars, 10 µm. (**C**) Quantification of relative intensities of E3-GFP and mCherry-LC3B on GUV membranes in (**A**) (means ± SDs are shown; N = 40). p≥0.5: (ns); 0.01<p<0.05: (*); 0.001<p<0.01: (**); p<0.001 (***); p<0.0001 (****).

likely have a dominant negative effect on autophagosome formation. Consistent with this observation, the R108E WIPI2 mutant was previously found to have a dominant negative effect on autophagosome formation in cells that were partially depleted in WIPI2 by RNAi knockdown (*Dooley et al., 2014*). WIPI2 with the K88E point mutation was able to facilitate autophagosome formation at a level that was not significantly different than Halo-tagged WT WIPI2, but not to the same extent as cells containing endogenous WIPI2 (p<0.0001). This is consistent with results shown in *Figure 4*, where the K88E mutation did not have as severe an impact on LC3B lipidation as the H85E and I92E mutations.

In addition to autophagosome number, we also analyzed the number of discrete WIPI2 puncta visualized with the Halo-tag (*Figure 5C*). We found significantly more WIPI2 puncta in cells transfected with WIPI2 that contained one of the three mutations shown to disrupt autophagosome formation compared to WT (H85E, p<0.0001; I92E, p<0.01; R108E, p<0.0001). Interestingly, the WIPI2 construct containing three mutations (H85E, I92E, and R108E) did not form significantly more puncta compared to WT WIPI2. This suggests that mutations that disrupt, but not eliminate, the W2IR interface may still recruit ATG16 but either fail to function efficiently or are improperly cleared.

Altogether, these data support the model that the interface between WIPI2 and ATG16 mediates autophagosome formation under starvation conditions. Notably, autophagosome formation persists at higher levels in cells without WIPI2 than is observed upon expression of some mutated constructs of WIPI2. Therefore, these data also suggest a robust mechanism for autophagosome formation in which WIPI2 may be one component of the preferential, but not sole, machinery with the potential to orchestrate autophagosome formation.

## In vitro reconstitution of WIPI2 membrane recruitment

We examined whether WIPI2 recruitment to GUV membranes was perturbed by the W2IR binding site mutations. A simple system including PI3KC3-C1 and E3 was used to explore the possibility that even in the presence of PI(3)P, E3 binding might contribute to WIPI2 recruitment. K88E, R108E, and R125E decreased WIPI2 recruitment to a significant extent (*Figure 6A,B*), while other mutants did not. In order to determine whether the loss in recruitment resulted from decreased interactions with E3 or membranes, an even simpler model was tested in which 5 % PI(3)P was included in the GUVs but no proteins other than WIPI2 were present (*Figure 6C*). K88E and K128E reduced binding to pure lipid membranes, but other mutants tested, including R108E and R125E, did not (*Figure 6C*). The effect of the K88E and K128E mutations on binding to PI(3)P-containing GUVs was unexpected, given that these residues are located distal to the FRRG motif involved in the known structural PI(3)P binding site (*Baskaran et al., 2012*; *Krick et al., 2012*), and suggests that membrane binding by WIPIs may be more complex than previously appreciated. The defects in autophagosome formation may thus represent a combination of defects in both lipid membrane and ATG16L1 binding. The unique effect of R108E and R125E on ATG16L1 binding and their strong autophagosome formation phenotype confirm the functional importance of ATG16L1 recruitment by WIPI2.

## Comparison across the WIPI protein family

The structure reported here was based on a construct corresponding to a consensus of the WIPI2b/d sequences for blades 1–7, since the C-terminal extension, the only region of divergence between the two proteins was deleted. These are the two WIPI2 isoforms that have been previously shown to bind ATG16L1 in immunoprecipitations from cells (*Dooley et al., 2014*). While the remaining WIPI isoforms diverge from the 2b/d consensus in blade 1, their sequences are identical in the blades 2 and 3 involved in ATG16L1 binding site. To the extent that these other isoforms were reported not to bind ATG16L1, these differences cannot be inherent in the W2IR binding groove itself, but rather must reflect other differences in the cellular context and modifications.

The only other human WIPI for which a structure is known is that of WIPI3 (*Liang et al., 2019*; *Ren et al., 2020*). WIPI3 interacts with the lipid transporter ATG2A (*Ren et al., 2020*) via what is believed to be a conserved binding site also present in WIPI4. WIPI4 is responsible for recruiting the phospholipid conduit ATG2A to sites of phagophore initiation, where it promotes tethering of the nascent

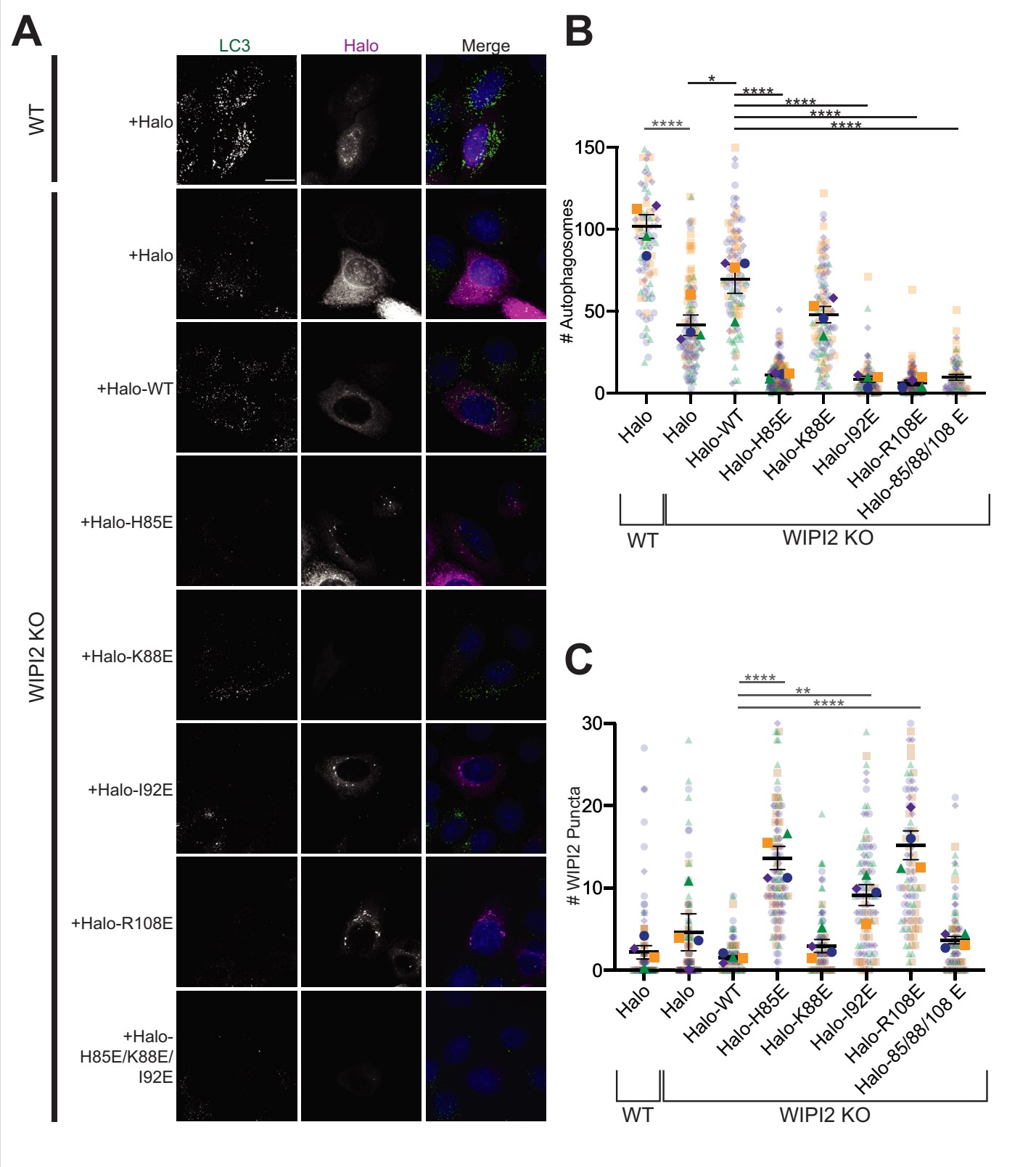

**Figure 5.** Altering the electrostatic interface of WIPI2 impairs starvation-induced autophagy in MEFs. (**A**) Representative maximum projections of LC3 staining and either Halo or Halo-WIPI2 signal in WT or WIPI2 knockout (KO) cells (indicated on left) following 2 hr of starvation and 100 nM BafA treatment. Scale bar 15 μm. (**B**) Number of LC3-positive autophagosomes in either WT cells transfected with Halo, WIPI2 KO cells transfected with Halo, or WIPI2 KO cells transfected with the indicated Halo-tagged WIPI2 construct (labeled by mutation). (**C**) Number of discrete WIPI2 puncta under the

*Figure 5 continued on next page*

*Figure 5 continued*

conditions described in (**B**), measured using maximum projections of Halo-tag fluorescence. Experimental replicates are color-coded, with translucent dots representing individual measurements from each replicate and opaque dots, the corresponding arithmetic mean of that replicate. Error bars ± SEM; n = 4 independent experiments; *p<0.05; **p<0.01; ***p<0.001; ****p<0.0001 based on a one-way ANOVA with Tukey's multiple comparisons between all conditions.

phagophore to the ER membrane source (*Chowdhury et al., 2018*; *Zheng et al., 2017*). The structure of WIPI3 is superimposable on that of WIPI2d with a Cα r.m.s.d. of 1.2 Å (*Figure 7A,B*).

## Comparison of yeast and human membrane recruitment of Atg16

A recently reported structure of the yeast WIPI2 ortholog *K. lactis* Atg21 (*Munzel et al., 2021*) bound to a fragment of yeast (*A. gossypii*) Atg16 allowed us to make a direct comparison of Atg16 membrane recruitment across species. Of the three yeast WIPI orthologs Atg18, Hsv2, and Atg21, it is Atg21 that recruits Atg12–5–16, the preautophagosomal membrane for Atg8 lipidation through

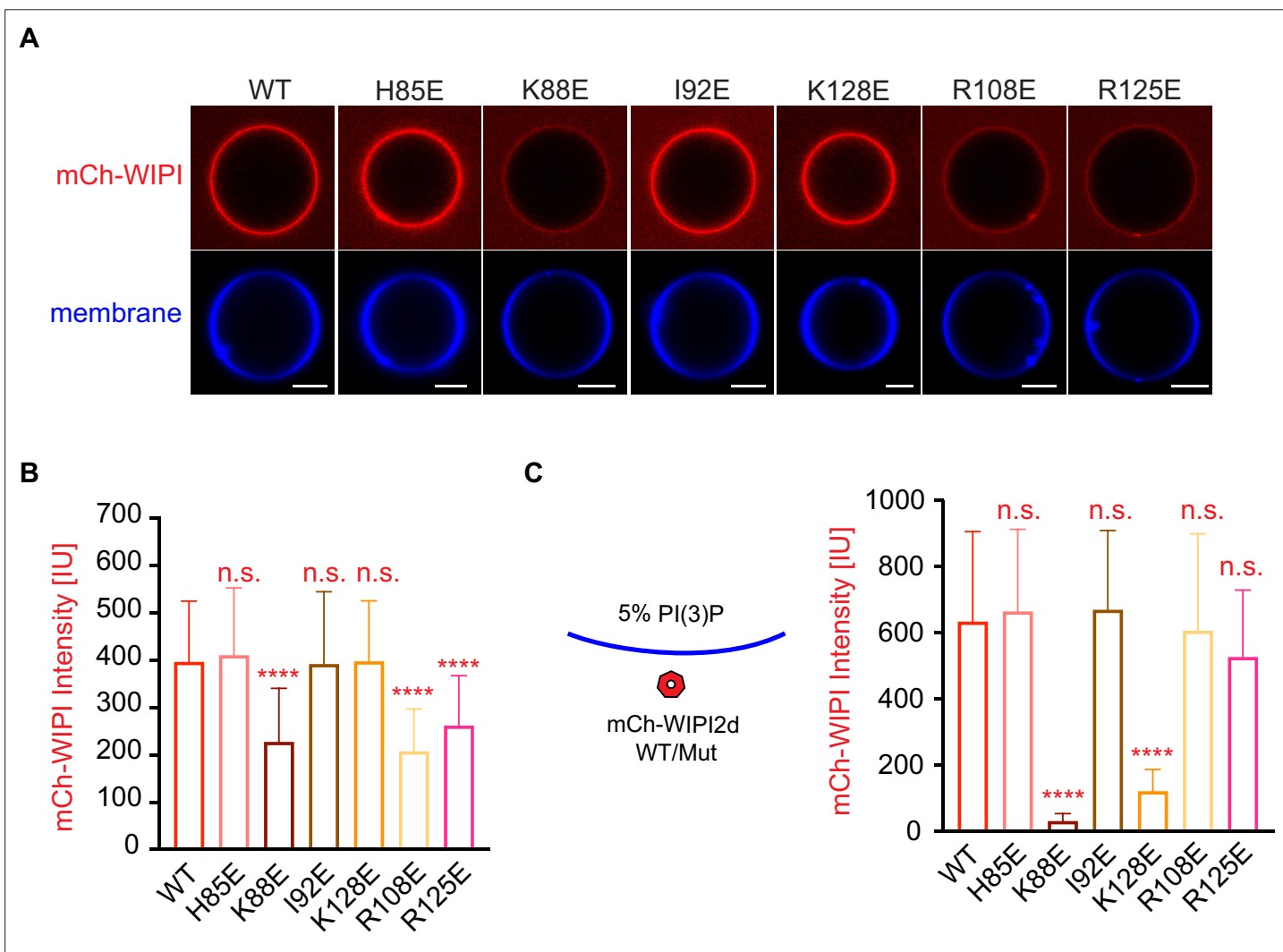

**Figure 6.** Reconstitution of membrane recruitment of WIPI2d mutants. (**A**) Representative confocal images of GUVs showing membrane binding of mCherry-WIPI2d. PI3KC3-C1, mCherry-WIPI2d WT or mutant, E3-GFP were incubated with GUVs (64.8 % DOPC: 20 % DOPE: 5 % DOPS: 10 % POPI: 0.2 % Atto647 DOPE) at room temperature. Images taken at 30 min were shown. Scale bars, 10 μm. (**B**) Quantification of relative intensities of mCherry-WIPI2d on GUV membranes in (**A**) membranes (means ± SDs are shown; N = 40). (**C**) Quantification of confocal images of GUVs (69.8 % DOPC: 20 % DOPE: 5 % DOPS: 5 % DOPI(3)P: 0.2 % Atto647 DOPE) showing membrane binding of mCherry-WIPI2d. mCherry-WIPI2d WT or mutant were incubated with GUVs at room temperature for 30 min and then imaged. (Means ± SDs are shown; N = 40). p≥0.5: (ns); 0.01<p<0.05: (*); 0.001<p<0.01: (**); p<0.001 (***); p<0.0001 (****).

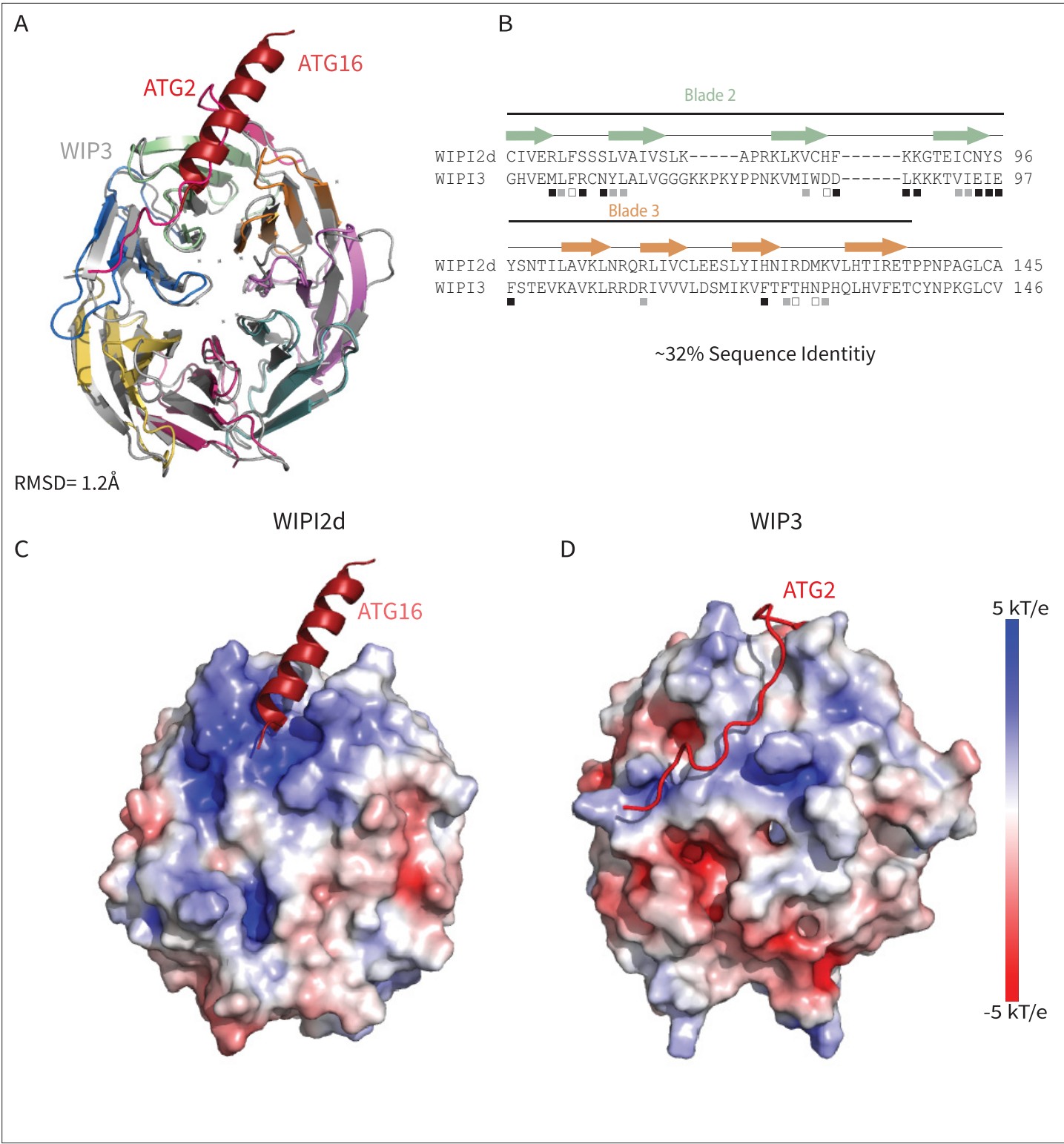

**Figure 7.** Comparing WIPI2d and WIPI3 structures and binding modes. Comparison of WIPI2d and WIPI3. Alignment of WIPI2d and WIPI3 (**A**) structure and (**B**) sequence based on structures with W2IR residues denoted with white squares, W34IR with black, and from both with gray. Electrostatic surface comparison of (**C**) WIPI2d and (**D**) WIPI3.

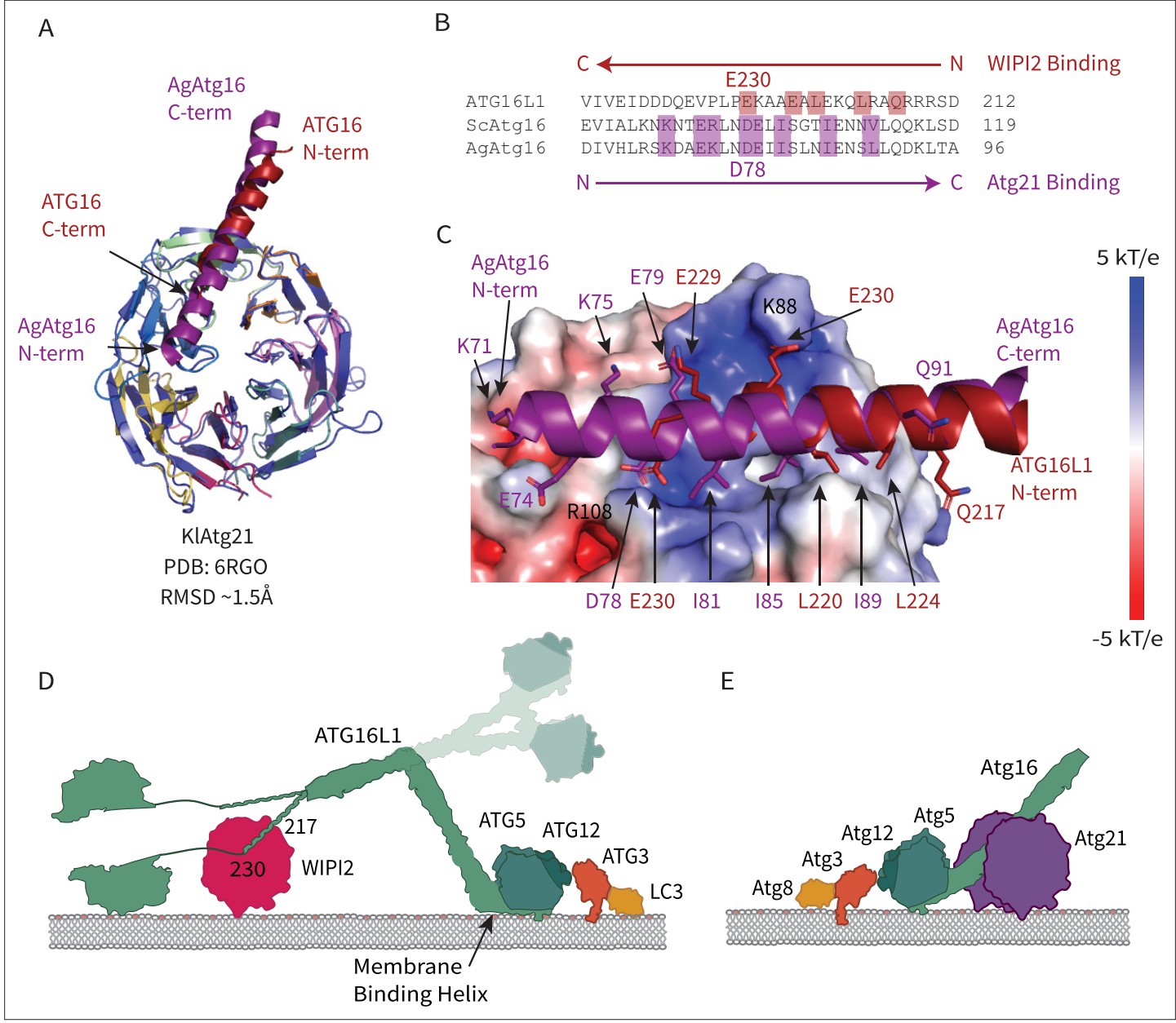

**Figure 8.** Comparison of WIPI2 and Atg21 binding to Atg16. (**A**) Structural alignment of WIPI2 (PDB: 7MU2) and Atg21 (PDB: 6RGO; indigo) structures bound to Atg16. (**B**) Sequence alignment of Atg16 ß-propeller binding residues based on structure. Residues for WIPI2 and Atg21 binding are in highlighted in red and purple, respectively. (**C**) Electrostatic potential of Atg21 with overlay of AgAtg16 and ATG16L1 in purple and red, respectively. Key interacting residues are shown in sticks and labeled. Model of ATG12–5-16 performing LC3 lipidation on the autophagic membrane with (**D**) WIPI2 recruitment in humans with Helix one membrane binding is labeled (***Lystad et al., 2019***), and a secondary upward conformation is shown in faded colors versus (**E**) Atg21 recruitment in yeast.

its interaction with Atg16 in yeast (***Juris et al., 2015***). The two ß-propeller domains of KlAtg21 and WIPI2d align well with a Cα r.m.s.d. of 1.5 Å, and both contain a basic patch on blade 2 (***Figure 8A,C***). The Atg16 peptide is bound between blades two and three in both structures (***Figure 8A***). Atg16 residues involved in binding share a similar composition (***Figure 8B,C***). Both AgAtg16 and ATG16L1 interact through a salt bridge situated in the 3AB loop of the ß-propeller and hydrophobic residues that are favorably buried between blades two and three. Remarkably, despite these similarities, the N-terminal to C-terminal orientation of the Atg16 helix is reversed (***Figure 8B,C***). Atg21 orients the N-terminus of AgAtg16 towards the membrane, while WIPI2 orients ATG16L1 N-terminus away from the membrane (***Figure 8D,E***).

## Discussion

WIPI2 is the linchpin of the circuit that connects two of the key reactions in autophagy initiation, the synthesis of PI(3)P by PI3KC3-C1, and LC3 lipidation by ATG12–5-16 L1. The WIPI2-ATG16L1 interaction is essential for starvation-induced bulk autophagy and xenophagy (*Dooley et al., 2014*) and for efficient LC3 lipidation in a reconstituted system with physiologically reasonable nanomolar concentrations of autophagy core complexes (*Fracchiolla et al., 2020*). From the perspective of therapeutic restoration of autophagic function in aging and neurodegeneration, ectopic expression of WIPI2b restores a normal rate of autophagosome biogenesis in aged neurons (*Stavoe et al., 2019*). Here, we report the high-resolution crystal structure of human WIPI2 and show how its unique electropositive and hydrophobic groove between blades 2 and 3 binds to the ATG16L1 W2IR.

The functional relevance of the groove residues was investigated by in vitro LC3 lipidation assays and by LC3 puncta formation in starvation-induced autophagy. All but one of the binding site mutants, K128E, reduced in vitro binding as judged by pull-down assays of purified proteins. WIPI2 activation of LC3 lipidation of GUV membranes by ATG12–5-16L1 precisely mirrored the results of the pull-down assays, with K128E again being the only mutant exhibiting no reduction. In cells, LC3 puncta formation was also reduced by most of the mutants, although the pattern did not follow the same rank order as the in vitro results. We interpret these data as confirmation that the W2IR binding site is important for LC3 lipidation in vivo, but that the many additional autophagy initiation components, including lipid membranes, present in cells still modulate the effects in subtle ways. Indeed, two of the mutants tested perturbed lipid membrane binding despite being distant from the known PI(3)P binding FRRG motif. In a simple linear paradigm of autophagy initiation, PI(3)P generated by PI3KC3-C1 recruits WIPI2, which in turn recruits E3 to catalyze LC3 lipidation. In this model, mutations that perturb the E3 binding of WIPI2 would not be expected to alter the recruitment of WIPI2 itself. However, at least one other upstream component, FIP200 (*Fujita et al., 2013*; *Gammoh et al., 2013*; *Nishimura et al., 2013*), contributes to E3 recruitment, and ATG16L1 has inherent membrane binding of its own (*Lystad et al., 2019*). Thus, the presence of E3 can stabilize WIPI2 on membranes in cells, a finding bolstered by our observation of the same effect in vitro.

Remarkably, the binding site for ATG2A is between blades 2 and 3 of WIPI3, the same two blades involved in binding ATG16L1 by WIPI2 (*Figure 7*). Despite the overall close similarity in the folds of the two WIPIs, the detailed structure of the blade 2–3 groove is quite divergent, explaining why WIPI3 does not bind ATG16L1, and WIPI2 does not bind to ATG2A. The Val- and Pro-rich ATG2A sequence that binds to WIPI3 in an extended conformation (*Ren et al., 2020*), and presumably WIPI4, is completely different in character from the Leu- and Glu-rich helical W2IR of ATG16L1. We propose the term WIPI3/4 interacting region (W34IR) for the ATG2A binding motif to contrast it with the distinct W2IR of ATG16L1. The ATG2A binding groove of WIPI3 is electrostatically neutral, as compared to the electropositive groove in WIPI2. A subset of the essential W2IR binding residues of WIPI2 (*Figure 7*, white squares) are altered in WIPI3. For example, the critical His 85 of WIPI2 is replaced by Asp in WIPI3. Expanding the analysis to WIPI1 and 4, the main features of the WIPI2 ATG16L1 binding groove are preserved in WIPI1, but not WIPI4 (*Figure 9*). Conversely, the ATG2A binding groove of WIPI3 is preserved in WIPI4, but not WIPI1 (*Figure 9*). Thus, the structural findings are consistent with the concept that the four human WIPIs can be subclassified into two groups (*Polson et al., 2010*): an ATG16L1-binding WIPI1/2 group and an ATG2A-binding WIPI3/4 group.

Whilst WIPI-based recruitment of ATG16L1 is critical for autophagy, a number of other factors are also involved. FIP200 can recruit ATG16L1 to sites of phagophore initiation (*Fujita et al., 2013*; *Gammoh et al., 2013*; *Nishimura et al., 2013*) via the central region of ATG16L1 that centers on residues 239–246 (*Fujita et al., 2013*) and so adjoins with the WIPI2 binding site. Binding to FIP200 alone in the absence of WIPI2 binding does not support autophagy induction (*Dooley et al., 2014*), and the nature of the interplay between FIP200 and WIPI2 binding to the ATG16L1 central region will be important to clarify. The Golgi-resident RAB33B also binds to ATG16L1 (*Itoh et al., 2008*), although the role of this interaction in autophagy is unclear. The RAB33B interaction was recently mapped structurally (*Metje-Sprink et al., 2020*), and the RAB33B binding site was found to terminate at ATG16L1 residue 210, just N-terminal to the first-ordered residues in the W2IR. In principle, it seems possible that RAB33B, FIP200, and WIPI2 might be capable of binding simultaneously.

Orienting WIPI2d membrane in the edge-on geometry proposed on the basis of previous studies (*Baskaran et al., 2012*; *Krick et al., 2012*), the N-terminus of the W2IR projects in the direction

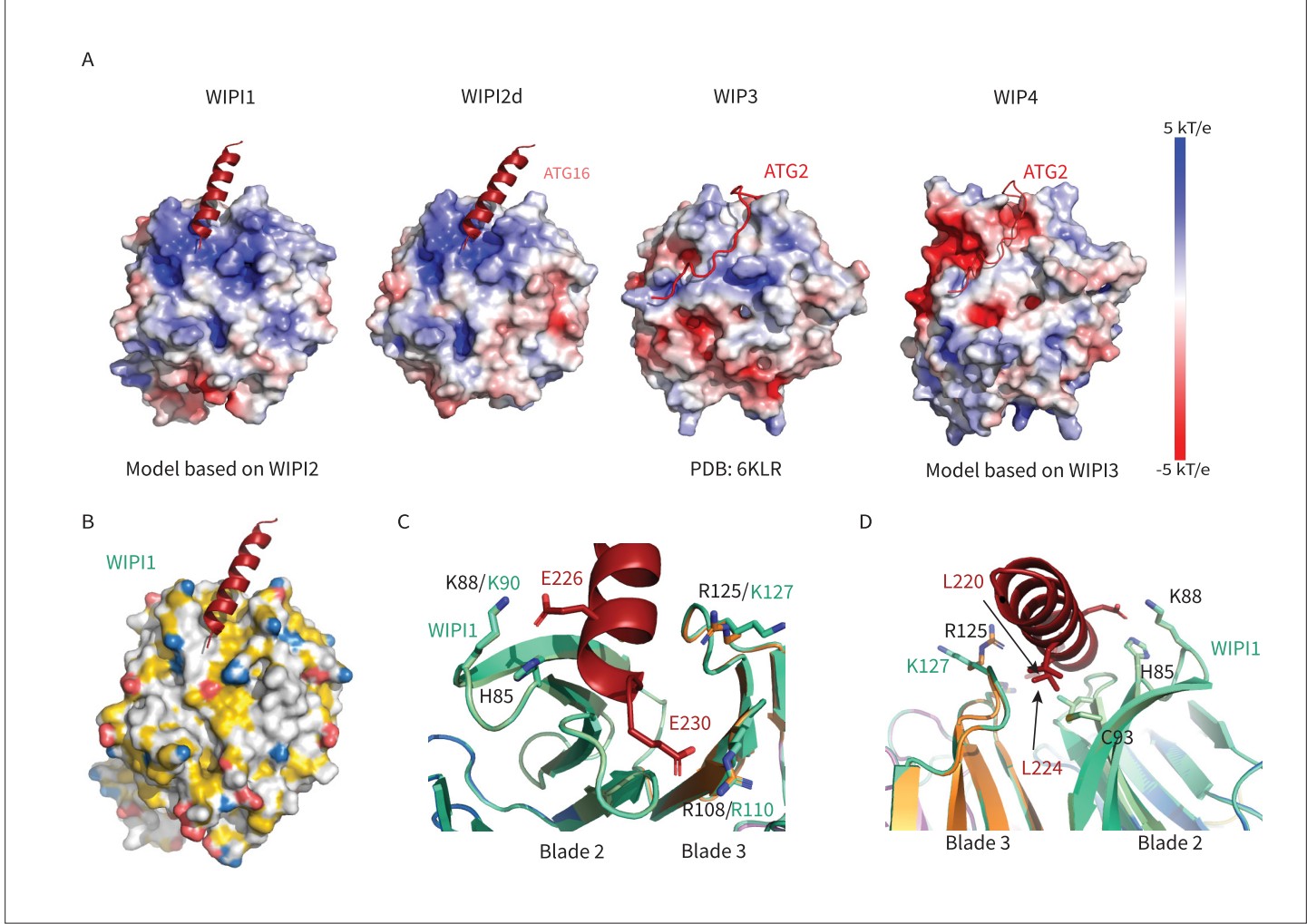

**Figure 9.** WIPI1-4 comparison. Comparison of electrostatic surface potential of (**A**) WIPI1-4. (**B**) Hydrophobic surface of WIPI1 with predicted ATG16L1 W2IR shown as cartoon. (**C, D**) Alignment of WIPI2d crystal structure and WIPI1 homology structure with WIPI1 shown as light green and key residues labeled in the same color as structure.

opposite to the membrane (*Figure 8D*). This potentially positions the ATG16L1 coiled coil to project away from the PI(3)P-containing membrane to which WIPI2 is bound. One model is that ATG16L1 could conjugate LC3 to the nascent phagophore in trans whilst anchored to a PI(3)P-containing domain of the ER (*Dooley et al., 2014*). In vitro, however, it is possible for WIPI2 to efficiently stimulate LC3 lipidation PI(3)P containing membranes in cis (*Fracchiolla et al., 2020*). On the basis of the recent yeast Atg21:Atg16 crystal structure (*Munzel et al., 2021*), and the presence of an intact yeast Atg21-binding motif at residues 164–165 of human ATG16L1, we predict that Atg21 is capable of binding to the E3 and positioning its active Atg12–Atg5 unit 'backwards' relative to its positioning by WIPI2. This likely explains why Atg21 is capable of targeting the human E3 to membranes yet fails to activate it for LC3 lipidation (*Fracchiolla et al., 2020*). Given the possibility that the ATG16L1 coiled coil can pivot with respect to the W2IR, these structural data on their own do not rule *cis* or *trans* LC3 lipidation in or out. Additional structures of ATG16L1 as assembled with multiple regulators, in the context of the full ATG12–5-16 L1 complex, and in the context of membranes, will be required to answer this question. The high-resolution structure presented here will be an important component for the interpretation of the larger scale, yet likely lower resolution, structures of assemblies yet to be solved.

## Materials and methods

**Key resources table**

| Reagent type (species) or resource | Designation | Source or reference | Identifiers | Additional information |
|---|---|---|---|---|
| Cell line (*Homo sapiens*) | HeLa human epithelial cell line | ATCC | CCL-2 | Authenticated by STR profiling; tested negative for mycoplasma |
| Cell line (*Homo sapiens*) | WIPI2-KO cells: HeLa cell line gene-edited to knockout WIPI2 expression | *Fischer et al., 2020* | | |
| Recombinant DNA reagent | pHTC HaloTag | Promega | G7711 | |
| Recombinant DNA reagent | Halo-WIPI2-WT (*Homo sapiens*) plasmid for transfection | *Stavoe et al., 2019* | Addgene 175025 | Available from Addgene |
| Recombinant DNA reagent | Halo-WIPI2-H85E (*Homo sapiens*) plasmid for transfection | Modified from Halo-WIPI2-WT in *Stavoe et al., 2019* | Addgene 175027 | Available from Addgene |
| Recombinant DNA reagent | Halo-WIPI2-K88E (*Homo sapiens*) plasmid for transfection | Modified from Halo-WIPI2-WT in *Stavoe et al., 2019* | Addgene 175028 | Available from Addgene |
| Recombinant DNA reagent | Halo-WIPI2-I92E(*Homo sapiens*) plasmid for transfection | Modified from Halo-WIPI2-WT in *Stavoe et al., 2019* | Addgene 175029 | Available from Addgene |
| Recombinant DNA reagent | Halo-WIPI2-R108E (*Homo sapiens*) plasmid for transfection | *Stavoe et al., 2019* | Addgene 176004 | Available from Addgene |
| Recombinant DNA reagent | Halo-WIPI2-H85/K88/I92E (*Homo sapiens*) plasmid for transfection | Modified from Halo-WIPI2-WT in *Stavoe et al., 2019* | Addgene 175033 | Available from Addgene |
| Antibody | Anti-LC3B, (Rabbit polyclonal) primary antibody | Abcam | Cat.#ab48394 | IF (1:1000) |
| Antibody | Anti-Rabbit AlexaFluor488, (Goat polyclonal) secondary antibody | ThermoFisher | Cat.#A11034 | IF (1:1000) |
| Chemical compound, drug | TMRDirect Halo Ligand | Promega | Cat.#G2991 | 37.5 nM final concentration |
| Software, algorithm | FIJI | PMID:22743772 | | |
| Software, algorithm | Ilastik | PMID:31570887 | | |
| Software, algorithm | Adobe Illustrator 2021 | Adobe | | |
| Software, algorithm | Prism 9 | GraphPad | | |
| Other | 35 mm #1.5 glass bottom imaging dishes | MatTek | Cat.# P35G-1.5–20 C | |
| Other | EBSS | ThermoFisher | Cat.# 24010043 | |
| Cell line (*Homo sapiens*) | HEK GnTi | ATCC | CRL-3022 | |
| Recombinant DNA reagent | pCAG-WIPI2d-cs-TEV | *Fracchiolla et al., 2020* | Addgene 171419 | |
| Recombinant DNA reagent | pCAG-WIPI2d10-364Δ263–295-cs-TEV | This paper | Addgene 171830 | Materials and methods section: Plasmids |
| Recombinant DNA reagent | pCAG-WIPI2dH85E-cs-TEV | This paper | Addgene 171831 | Materials and methods section: Plasmids |
| Recombinant DNA reagent | pCAG-WIPI2dK88E-cs-TEV | This paper | Addgene 171832 | Materials and methods section: Plasmids |
| Recombinant DNA reagent | pCAG-WIPI2dI92E-cs-TEV | This paper | Addgene 171833 | Materials and methods section: Plasmids |
| Recombinant DNA reagent | pCAG-WIPI2dC93E-cs-TEV | This paper | Addgene 171834 | Materials and methods section: Plasmids |
| Recombinant DNA reagent | pCAG-WIPI2dR108E-cs-TEV | This paper | Addgene 171835 | Materials and methods section: Plasmids |

| Reagent type (species) or resource | Designation | Source or reference | Identifiers | Additional information |
|---|---|---|---|---|
| Recombinant DNA reagent | pCAG-WIPI2dR125E-cs-TEV | This paper | Addgene 171836 | Materials and methods section: Plasmids |
| Recombinant DNA reagent | pCAG-WIPI2dK128E-cs-TEV | This paper | Addgene 171837 | Materials and methods section: Plasmids |
| Recombinant DNA reagent | pCAG-mcherry-WIPI2d-cs-TEV | *Fracchiolla et al., 2020* | Addgene 178912 | |
| Recombinant DNA reagent | pCAG-mcherry-WIPI2dH85E-cs-TEV | This paper | Addgene 171838 | Materials and methods section: Plasmids |
| Recombinant DNA reagent | pCAG-mcherry-WIPI2dK88E-cs-TEV | This paper | Addgene 171839 | Materials and methods section: Plasmids |
| Recombinant DNA reagent | pCAG-mcherry-WIPI2dI92E-cs-TEV | This paper | Addgene 171840 | Materials and methods section: Plasmids |
| Recombinant DNA reagent | pCAG-mcherry-WIPI2dC93E-cs-TEV | This paper | Addgene 171841 | Materials and methods section: Plasmids |
| Recombinant DNA reagent | pCAG-mcherry-WIPI2dR108E-cs-TEV | This paper | Addgene 171842 | Materials and methods section: Plasmids |
| Recombinant DNA reagent | pCAG-mcherry-WIPI2dR125E-cs-TEV | This paper | Addgene 171843 | Materials and methods section: Plasmids |
| Recombinant DNA reagent | pCAG-mcherry-WIPI2dK128E-cs-TEV | This paper | Addgene 171844 | Materials and methods section: Plasmids |
| Recombinant DNA reagent | pLEXm-GST-TEV-ATG14 | | Addgene 99329 | |
| Recombinant DNA reagent | pCAG-TSF-TEV-BECN1 | | Addgene 99328 | |
| Recombinant DNA reagent | pCAG-TSF-TEV-VPS34 | | Addgene 99327 | |
| Recombinant DNA reagent | pCAG-VPS15 | *Stjepanovic et al., 2017* | Addgene 99326 | |
| Recombinant DNA reagent | pGBdest-ATG12-10xHis-TEV-ATG5-10xHis-TEVcs-ATG16L1-GFP-TEVcs-StrepII, ATG7, ATG10 | *Fracchiolla et al., 2020* | Addgene 169077 | |
| Recombinant DNA reagent | pFast BacHT(B)–6xHis-TEV-ATG7 | *Fracchiolla et al., 2020* | | |
| Recombinant DNA reagent | pET Duet-1-6xHis-TEV-ATG3 | *Fracchiolla et al., 2020* | Addgene 169079 | |
| Recombinant DNA reagent | pET Duet-1-6xHis-TEV-mCherry-LC3B-Gly(Δ5 C) | *Zaffagnini et al., 2018* | Addgene 169168 | |
| Other | 96–2 well,INTELLI-PLATE (original) tray | Molecular Dimensions, Maumee, OH | | |
| Other | Greiner pre-greased 24 well Combo Plate (SBS format) with lid | Molecular Dimensions, Maumee, OH | | |
| Software, algorithm | Nikon Elements microscope imaging software 4.60 | Nikon Corporation, Tokyo, Japan | https://www.nikoninstruments.com/Products/Software/NIS-Elements-Advanced-Research/NIS-Elements-Viewer | |
| Other | Glutathione Sepharose 4B GST-tagged protein purification resin | GE healthcare, Chicago, IL | Cat# 17075605 | |
| Other | Strep-Tactin Superflow high capacity 50 % suspension | IBA Lifesciences,Göttingen, Germany | Cat# 2-1208-010 | |
| Software, algorithm | phenix.refine | PMID:20124702, 22505256, 31588918 | RRID:SCR_016736 | |
| Software, algorithm | XDS | PMID:20124692 | RRID:SCR_015652 | |

| Reagent type (species) or resource | Designation | Source or reference | Identifiers | Additional information |
|---|---|---|---|---|
| Software, algorithm | POINTLESS | PMID:21460446 | RRID:SCR_014218 | |
| Software, algorithm | PHASER | PMID:19461840 | RRID:SCR_014219 | |
| Software, algorithm | PyMol | PyMol (pymol.org) | RRID:SCR_000305 | |
| Software, algorithm | APBS | PMID:11517324 | RRID:SCR_008387 | |
| Software, algorithm | SWISS-MODEL | PMID:29788355 | RRID:SCR_018123 | |
| Software, algorithm | Coot | PMID:20383002 | RRID:SCR_014222 | |

## Plasmids

WIPI2d crystallography constructs and mutants were sub-cloned from a plasmid from a previous study (*Fracchiolla et al., 2020*) into the pCAG vector using restriction enzyme cloning. mCherry constructs were cloned similarly with an N-terminal mCherry tag. All constructs had a C-terminal TEV cleavage site followed by TwinStrep tags. WIPI2b WT and R108E plasmids for mammalian cell transfection were as previously described (*Stavoe et al., 2019*); additional point mutations H85E, K88E, I92E, and the triple mutation 85/88/108E were introduced into the WT construct and verified by sequencing. Protocol available at https://doi.org/10.17504/protocols.io.bxktpkwn (*Strong, 2021a*).

## Protein expression and purification

Purification of WIPI2d constructs used for crystallization, pull-down assays, and GUV assays were expressed in HEK GnTi cells. Constructs were transfected to cells using polyethylenimine (Poly-sciences). After 60 hr of expression, cells were harvested and lysed with lysis buffer (50 mM Hepes, pH 7.4, 1 % Triton X-100, 300 mM NaCl, and 1 mM Tris(2-carboxyethyl)phosphine [TCEP]) supplemented with EDTA-free protease inhibitors (Roche). The lysate was clarified by centrifugation (17,000 rpm for 1 hr at 4 °C) and incubated with StrepTactin Sepharose resin (IBA) for 2 hr at 4 °C, applied to a gravity column, and washed extensively with wash buffer (50 mM Hepes, pH 7.4, 300 mM NaCl, and 1 mM TCEP). The protein complexes were eluted with wash buffer containing 10 mM desthiobiotin (Sigma) and treated with TEV protease at 4 °C overnight. Cleaved protein was applied to a Superdex 200 column (16/60 prep grade) equilibrated with gel filtration buffer (25 mM Hepes, pH 7.4, 150 mM NaCl, and 1 mM TCEP). Peak fractions were collected, pooled, snap frozen in liquid nitrogen, and stored at –80 °C. Purification of ATG12–5-16, PI3KC3-C1, ATG7, ATG3, and LC3 used for GUV assays were performed as previously described (*Fracchiolla et al., 2020*). Protocols are available at https://doi.org/10.17504/protocols.io.buxqnxmw (*Strong, 2021b*), https://doi.org/10.17504/protocols.io.br6qm9dw (*Fracchiolla, 2021a*), https://doi.org/10.17504/protocols.io.bseenbbe (*Chang, 2021a*), https://doi.org/10.17504/protocols.io.bsennbde (*Fracchiolla, 2021b*), https://doi.org/10.17504/protocols.io.btgknjuw (*Turco and Fracchiolla, 2021*), https://doi.org/10.17504/protocols.io.btiunkew (*Fracchiolla, 2021c*).

## Crystallization and structural determination

WIPI2d10-364Δ263–295: ATG16L1 (207–230) complex was formed overnight with 5 × molar excess peptide (GenScript). Crystals of the complex were grown using hanging drop vapor diffusion method at 4 °C. One μL of the protein complex (2 mg/mL) was mixed with one μL reservoir solution and 0.3 μL of a crystal seed stock. This was suspended over a 500 μL reservoir of 22% w/v PEG 3,350 (Hampton Research), 2% v/v Tacsimate pH 7.0 (Molecular Dimension), and 100 mM Hepes pH 7.7. Crystals appeared within 2 days and were continued to grow for approximately a week. Crystals were cryoprotected in reservoir solution supplemented with 25 % (v/v) glycerol. A native dataset was collected from a single crystal under cryogenic conditions (100 K) at a wavelength of 0.979 Å using a Dectris PILATUS 6 M/EIGER 16 M detector (beamline BL12-2, SSRL). The data was indexed and integrated using LABELIT and XDS (*Kabsch, 2010*). Integrated reflections were scaled, merged, and truncated using AIMLESS and TRUNCATE, respectively. Initial phases were determined by molecular replacement with the program PHASER (*McCoy et al., 2007*) using KIHsv2 (PDB: 4EXV) (*Baskaran et al., 2012*) as a search model. ATG16L1 peptide was manually modeled into the structure according to the 2Fo-Fc and Fo-Fc electron density maps using Coot (*Emsley et al., 2010*). Iterative rounds of

manual model building and refinement were performed using Coot (*Emsley et al., 2010*) and Phenix Refine (*Afonine et al., 2012*), respectively (https://www.mrc-lmb.cam.ac.uk/public/xtal/doc/phenix/tutorials/mr_refine.html). Data collection and refinement statistics are listed in *Supplementary file 1*. WIPI2 ATG16L1 interface was analyzed using PDBePISA (*Krissinel and Henrick, 2007*). All figures were generated with PyMol (http://www.pymol.org). The electrostatic surface was calculated using APBS (*Baker et al., 2001*) https://github.com/Electrostatics/electrostatics.github.io; *Nathan, 2021*, in PyMOL. Hydrophobic surface was generated using YBR script in PyMOL (*Hagemans et al., 2015*). WIPI1 and WIPI4 homology models were generated in SWISS-Model (*Bertoni et al., 2017*; *Bienert et al., 2017*; *Studer et al., 2020*; *Studer et al., 2021*; *Waterhouse et al., 2018*) using WIPI2d10-364Δ263–295 and WIPI3 (PDB: 6KLR) as templates, respectively. Protocol available at https://doi.org/10.17504/protocols.io.bu7tnznn (*Strong, 2021c*).

## Coprecipitation assay

Ten micromolar  purified WIPI2d was mixed with 20 µM of GST or GST-ATG16L1(207–230) and 10 µL Glutathione Sepharose 4B (GE Healthcare). The final buffer was 25 mM HEPES pH 7.4, 150 mM NaCl, 1 mM TCEP. The final volume was 150 µL. The system was gently rocked at 4 °C for 2 hr before washing the protein-bound resin three times. Loading dye was added to the beads and bands were visualized using SDS–PAGE gel after coomassie staining. Three replicates were performed. Protocol available at https://doi.org/10.17504/protocols.io.bxkspkwe (*Strong, 2021d*).

## GUV assay

GUVs were prepared by hydrogel-assisted swelling as described previously (*Chang et al., 2021b*). The reactions were set up in an eight-well observation chamber (Lab Tek) that pre-coated with 5 mg/mL β casein for 30 min. For E3 membrane recruitment and LC3 lipidation assay, a final concentration of 50 nM PI3KC3-C1 complex, 250 nM WIPI2d or mutant proteins, 50 nM E3-GFP complex, 100 nM ATG7, 100 nM ATG3, 500 nM mCherry-LC3B, 50 µM ATP, and 2 mM MnCl2 were used. For WIPI2d membrane binding assay, a final concentration of 50 nM PI3KC3-C1, 400 nM mCherry-WIPI2d or mutant proteins, and 50 nM E3-GFP complex were used. A final volume of 120 µL mixture was made for all the reactions. Ten microliters  GUVs were added to initiate the reaction. After 5 min incubation, during which random views were picked for imaging, time-lapse images were acquired in multi-tracking mode on a Nikon A1 confocal microscope with a 63× Plan Apochromat 1.4 NA objective. Three biological replicates were performed for each experimental condition. Identical laser power and gain settings were used during the course of all conditions.

For quantification of protein intensity on GUV membranes, the outline of individual vesicle was manually defined based on the membrane channel. The intensity threshold was calculated by the average intensities of pixels inside and outside of the bead and then intensity measurements of individual bead were obtained. Averages and standard deviations were calculated among the measured values per each condition and plotted in a bar graph. The data were analyzed with GraphPad Prism nine by using one-way ANOVA with Dunn's multiple comparisons test. Protocol available at https://doi.org/10.17504/protocols.io.bxm2pk8e (*Chang, 2021b*).

## Starvation experiments in WT and WIPI2 knockout HeLa cells

WIPI2 knockout (KO) HeLa cells and their corresponding parent line were generously provided by Richard Youle (National Institute of Health). Cells were cultured in DMEM (10 % FBS, 1 % Pen/Strep, 1 % GlutaMAX). Cells were authenicated by STR profiling and tested as myoplasma-free at the Penn Genomic Analysis Core. Cells were transfected with 0.75 µg of the indicated WIPI2 construct or soluble Halo-tag control 18 hr prior to starvation using FuGENE transfection reagent as recommended. To induce starvation, cells were washed twice in 1 × Earle's balanced salt solution (EBSS) and incubated for 2 hr in EBSS containing 100 nM baflomycin A and TMR Direct Halo Ligand. To visualize autophagosomes, cells were fixed in ice-cold MeOH at −20 °C for 10 min. Cells were incubated in blocking solution (5 % normal goat serum, 1 % BSA, 0.05 % NaN3 in 1 × PBS) for 1.5 hr. Primary LC3 antibody (ab48394, 1 µg/mL) was diluted in blocking solution and used for 1 hr at room temperature. Cells were washed thrice in 1 × PBS and incubated in AlexaFluor 488 (1:1000 in blocking buffer). Cells were then washed once in 1 × PBS, incubated in PBS with Hoechst (4 µg/mL) for 10 min to allow for visualization of nuclei, washed thrice more, and stored at 4 °C. HeLa cells were imaged in PBS on a Perkin Elmer

spinning disk confocal setup with a Nikon Eclipse Ti inverted microscope, a Hamamatsu EMCCD 9100–50 camera, and an Apochromat 100 × 1.49 NA oil immersion objective. Images were acquired as z-stacks with a 200 nm step size.

Z-stacks were assembled into maximum projections and channels were split using FIJI (NIH). At least one image from each condition, compiled from across biological replicates (a unique passage of HeLa cells was considered a biological replicate), was used to train Ilastik to identify LC3 and WIPI2 puncta. Training images were not used in subsequent data analysis. Images from each experiment and for each condition were processed in batch mode by Ilastik to yield simple segmentation files. Using the Halo-tag channel, cell outlines were drawn by hand and saved as ROIs in FIJI. LC3 and WIPI2 puncta were counted within resulting ROIs using Analyze Particles in FIJI. For both LC3 and WIPI2 puncta, size was set to 0-Infinity (square pixels). Results were tabulated in Microsoft Excel; graphing and statistical tests were performed using GraphPad Prism 9. Superplots were generated as discussed in *Lord et al., 2020*. One-way ANOVAs were performed on the averages for the biological replicates; Tukey's multiple comparisons test was used post hoc to compare all conditions to each other. Protocol available at https://doi.org/10.17504/protocols.io.bxpdpmi6 (*Riley and Holzbaur, 2021*).

## Acknowledgements

We thank members of the Hurley and Holzbaur labs, Sascha Martens, Michael Lazarou and members of their laboratories, Dorotea Fracchiolla, and others in Aligning Science Across Parkinson's Team mito911 for advice and discussions. We thank Richard Youle for the WIPI2 knockout cell line, Clyde Smith and Lisa Dunn at SSRL beamline BL12-2 for assistance with data collection. The study is funded by the joint efforts of the Michael J Fox Foundation for Parkinson's Research (MJFF) and Aligning Science Across Parkinson's (ASAP) initiative. MJFF administers the grant ASAP-000350 (to JHH and EH) on behalf of ASAP and itself. CAB was supported by the German Research Foundation (DFG; BO 5434/1–1). The research was also supported by National Institute of General Medical Sciences, NIH, R01 GM111730 (JHH) and National Institute of Neurological Disease and Stroke, NIH, R00 NS109286 (AKHS). Use of the Stanford Synchrotron Radiation Lightsource, SLAC National Accelerator Laboratory, is supported by the U.S. Department of Energy, Office of Science, Office of Basic Energy Sciences under Contract No. DE-AC02-76SF00515. The SSRL Structural Molecular Biology Program is supported by the DOE Office of Biological and Environmental Research and by the National Institutes of Health, National Institute of General Medical Sciences (P30GM133894). The contents of this publication are solely the responsibility of the authors and do not necessarily represent the official views of NIGMS or NIH.

## Additional information

### Competing interests

James H Hurley: JHH is employed by UC Berkeley. JHH has a competing interest as a co-founder of Casma Therapeutics.. The other authors declare that no competing interests exist.

### Funding

| Funder | Grant reference number | Author |
| --- | --- | --- |
| Aligning Science Across Parkinson's | ASAP-000350 | Erika LF Holzbaur James H Hurley |
| National Institute of General Medical Sciences | R01 GM111730 | James H Hurley |
| National Institute of Neurological Disorders and Stroke | R00 NS109286 | Andrea KH Stavoe |

The funders had no role in study design, data collection and interpretation, or the decision to submit the work for publication.

## Author contributions
Lisa M Strong, Conceptualization, Data curation, Formal analysis, Investigation, Validation, Visualization, Writing - original draft; Chunmei Chang, C Alexander Boecker, Formal analysis, Investigation, Writing – review and editing; Julia F Riley, Andrea KH Stavoe, Investigation; Thomas G Flower, Xuefeng Ren, Investigation, Writing – review and editing; Cosmo Z Buffalo, Formal analysis, Investigation, Validation, Writing – review and editing; Erika LF Holzbaur, Funding acquisition, Supervision, Writing – review and editing; James H Hurley, Conceptualization, Formal analysis, Funding acquisition, Project administration, Supervision, Writing - original draft, Writing – review and editing

## Author ORCIDs
Lisa M Strong ⓘ http://orcid.org/0000-0002-4293-8131
Chunmei Chang ⓘ http://orcid.org/0000-0002-5607-7985
Julia F Riley ⓘ http://orcid.org/0000-0001-8518-9786
C Alexander Boecker ⓘ http://orcid.org/0000-0001-9701-5273
Thomas G Flower ⓘ http://orcid.org/0000-0002-7890-6473
Andrea KH Stavoe ⓘ http://orcid.org/0000-0002-4073-4565
Erika LF Holzbaur ⓘ http://orcid.org/0000-0001-5389-4114
James H Hurley ⓘ http://orcid.org/0000-0001-5054-5445

## Decision letter and Author response
Decision letter https://doi.org/10.7554/eLife.70372.sa1
Author response https://doi.org/10.7554/eLife.70372.sa2

## Additional files

### Supplementary files
• Supplementary file 1. Data collection and refinement statistics.
• Supplementary file 2. Oligos used for cloning.
• Transparent reporting form

### Data availability
Coordinates and structure factors have been deposited in the Protein Data Bank under accession code PDB 7MU2. Protocols have been deposited in protocols.io. Plasmids developed for this study will be deposited at Addgene.org. GUV source data have been deposited in Zenodo.

The following dataset was generated:

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
