## [Decision Letter]

**Acceptance summary:**

This paper describes the structure of WIPI2d in association with a region of ATG16L1 that interacts with WIPI. Previous work has demonstrated that WIPI proteins bind to PI3P on the surface of phagophores to initiate autophagy. WIPI also associates with ATG16 orthologs to template assembly of the phagophore expansion machinery. The authors map the interface, and also perform biochemical and cell biological experiments that support the molecular structure. Analysis of mutations within the interaction interface reveal the residues required for interaction, confirming the structural data. This information will be very important in efforts to biochemically reconstitute the initial stages of autophagy. This work also highlights the complications of examining mutations in complex interaction surfaces in cells, particularly in cases where multivalent interactions drive the process. The results also explain why there are multiple WIPI proteins in the human genome, reflecting distinct mechanisms of recruitment of ATG16 and ATG2 orthologs to the appropriate membrane.

**Decision letter after peer review:**

Thank you for submitting your article "Structural Basis for Membrane Recruitment of ATG16L1 by WIPI2 in Autophagy" for consideration by *eLife*. Your article has been reviewed by 3 peer reviewers, one of whom is a member of our Board of Reviewing Editors, and the evaluation has been overseen by Volker Dötsch as the Senior Editor. The reviewers have opted to remain anonymous.

Essential revisions:

Overall, the reviewers feel that the paper has merit and in particular provides structural clarity to WIPI2 and its various interactions. However, the reviewers feel that the paper needs additional experiments to fully substantiate the major conclusions. These fall into two areas:

1) As pointed out in detail by reviewer 3, additional data is needed on the binding of some of the WIPI2 mutants to membranes (GUVs) and in particular, addressing the question of whether binding of the mutants requires E3 or not. Specifically, the authors should test the recruitment to GUVs in the absence of E3 for the whole panel shown in Figure 6 (which is apparently done in the presence of E3).

2) Reviewers 1 and 2 commented on the functional studies performed in cells. In these experiments, siRNA was used to reduce the levels of WIPI2, although there was substantial endogenous protein remaining. This creates limitations in the conclusions that can be made in part because certain mutant proteins could be behaving in a dominant negative manner. The reviewers feel, after significant discussion, that the best way to address the function of the mutant WIPIs is to create a knockout allele and perform rescue experiments by expressing the mutant proteins at near endogenous levels. Otherwise one is left trying to argue loss of activity rather than a dominant negative effect. This could potentially be done on key mutants, and would strengthen the paper substantially.

3) It will be important to appropriately cite prior work from the Tooze lab, as mentioned by reviewer 2.

*Reviewer #1 (Recommendations for the authors):*

This paper describes the structure of WIPI2d in association with a region of ATG16L1 that interacts with WIPI. Previous work has demonstrated that WIPI proteins bind to PI3P on the surface of phagophores to initiate autophagy. WIPI also associates with ATG16 orthologs to template assembly of the phagophore expansion machinery. The authors map the interface, and also perform biochemical and cell biological experiments that support the molecular structure.

In detail, the authors employed a composite WIPI2b/WIPI2d consensus protein to facilitate crystallization and were able to resolve to <2.0 angstroms. The structure itself is straightforward and the presentation is clear. Mutations in the interface reduce binding between ATG16L1 and WIPI2d, as predicted by the structure.

The biochemical experiments looking at recruitment to GUVs are straightforward. To examine the relevance of the WIPI2-ATG16L1 interface in cells, the authors partially depleted WIPI2 and then rescued with point mutants or WT using Halo-fused WIPI. They do not show how much the WIPI rescue constructs are over expressed and this could impact some of the results in that at endogenous levels, there could be a stronger phenotype. Another limitation of the experimental approach is that WIPI1, WIPI3 and WIPI4 are still present and could serve partially redundant functions in terms of LC3 puncta. It isn't clear why the authors did not employ a more rigorous approach, such as endogenous knock-ins of the relevant mutants in order to address the question.

Although not surprising on a complex interface involving a WD40 surface, some of the mutants affect binding to PI3P on GUVs as well as binding to ATG16L1. The issue here is that some of the effects seen in cells could reflect mostly the PI3P binding function than the ATG16L1 binding function. Indeed the mutants with the strongest phenotypes for LC3 puncta also have the most drastic effect on PI3P binding. The authors address this in the discussion by essentially saying there are additional components that may work together in a partially redundant manner.

Finally, the authors end with some comparative analyses that explain much about why there are multiple WIPI proteins.

Overall, the strength of the paper is the strong biochemistry and crystallography. The in cellulo data largely support the model but are limiting in impact because of the system (RNAi) used and the other complex redundant pathways. In the end, it may be that biochemical reconstituted of the autophagy system provides a better understanding of the pathway than the in cellulo studies, from a mechanistic point of view, given the difficulty in interpreting redundant pathways.

*Reviewer #2 (Recommendations for the authors):*

In vivo confirmation of mutant relevance – The hypothesis/design of this figure is not well articulated in the text. It would be helpful to express the expectation that the mutants will function as dominant negatives and to cite that expectation with respect to the Tooze paper (assuming that is what the authors predict). It seems imperative to show what the LC3 puncta formation looks like in the knockdown alone. One interpretation could be that the mutants are all rescuing just not quite to the level of the wildtype protein, if the knockdown is already severely impaired. Furthermore, it seems likely that a more complete knockdown would provide some dynamic range to the LC3 results, making it easier to interpret whether these mutations are significantly impactful.

In Panel D, the most impressive result is with C93E, but in the in vitro experiments the authors note that this protein is largely insoluble. Is there any reason to believe it is better behaved in cells? If not, this is a simple explanation for why WIPI2 puncta numbers are reduced and further argues against showing any C93E data.

Is there a good reason to not show ATG16 puncta formation? This seems like the most direct test of the authors predictions, that these mutants will disrupt recruitment of the ATG16L1 complex?

Do the mutants in the ATG16L1 interaction site impact WIPI-membrane interaction? The authors interpret the reduction of some WIPI2 puncta in cells as evidence for a disruption in membrane binding and further, they suspect this is due to a cooperative interaction of WIPI with ATG16L1 and with membranes. To test this, they look at WIPI recruitment in their in vitro GUV assay in the presence of ATG16L1. It seems essential to do the same experiment without ATG16L1 though, to rule out the alternative interpretation that these mutations are having effects outside of ATG16L1 binding and perhaps are directly modifying the affinity of WIPI2 for phosphoinositide-rich membranes.

Figure 3 – D is not indicated in legend. The text describes the outcomes of this experiment for individual mutations as they bind with similar or low "affinity", but the design of the experiment does not avail itself to this metric. Furthermore, no replicates are shown or quantification provided.

Figure 4 and Figure 6, axes labeled "Indensity".

Figure 3 supplement (uncropped source data not included in main manuscript).

---

## [Author Response]

Essential revisions:Overall, the reviewers feel that the paper has merit and in particular provides structural clarity to WIPI2 and its various interactions. However, the reviewers feel that the paper needs additional experiments to fully substantiate the major conclusions. These fall into two areas:1) As pointed out in detail by reviewer 3, additional data is needed on the binding of some of the WIPI2 mutants to membranes (GUVs) and in particular, addressing the question of whether binding of the mutants requires E3 or not. Specifically, the authors should test the recruitment to GUVs in the absence of E3 for the whole panel shown in Figure 6 (which is apparently done in the presence of E3).

We thank the reviewers for this feedback. The suggested experiment was performed and added into Figure 6 as a new panel, C. Two mutants, K88E and K128E, do in fact perturb lipid membrane binding, despite being far from the FRRG motif/PI(3P) binding site.The membrane binding modes of WIPIs may thus be somewhat more complicated than previously appreciated. Other key ATG16L1 binding site mutants, including R108E and R125E, have no significant effect on recruitment in the absence of ATG16L1.

2) Reviewers 1 and 2 commented on the functional studies performed in cells. In these experiments, siRNA was used to reduce the levels of WIPI2, although there was substantial endogenous protein remaining. This creates limitations in the conclusions that can be made in part because certain mutant proteins could be behaving in a dominant negative manner. The reviewers feel, after significant discussion, that the best way to address the function of the mutant WIPIs is to create a knockout allele and perform rescue experiments by expressing the mutant proteins at near endogenous levels. Otherwise one is left trying to argue loss of activity rather than a dominant negative effect. This could potentially be done on key mutants, and would strengthen the paper substantially.

We redid the experiments using a WIPI2 knockout cell line generated by the Richard Youle’s lab and kindly provided to us by him. The loss of autophagosome formation phenotypes are much stronger, as expected. The previous experiments in the partial knockdown cells have been removed.

3) It will be important to appropriately cite prior work from the Tooze lab, as mentioned by reviewer 2.

We believe the reviewer was referring to the dominant negative experiments in the context of the knockdown experiments, which we have now removed. We do see a dominant negative effect of the mutants even in the context of the knockout, and mention that Dooley et al. and Tooze had seen this in the knockdown.

Reviewer #2 (Recommendations for the authors):In vivo confirmation of mutant relevance – The hypothesis/design of this figure is not well articulated in the text. It would be helpful to express the expectation that the mutants will function as dominant negatives and to cite that expectation with respect to the Tooze paper (assuming that is what the authors predict). It seems imperative to show what the LC3 puncta formation looks like in the knockdown alone. One interpretation could be that the mutants are all rescuing just not quite to the level of the wildtype protein, if the knockdown is already severely impaired. Furthermore, it seems likely that a more complete knockdown would provide some dynamic range to the LC3 results, making it easier to interpret whether these mutations are significantly impactful.In Panel D, the most impressive result is with C93E, but in the in vitro experiments the authors note that this protein is largely insoluble. Is there any reason to believe it is better behaved in cells? If not, this is a simple explanation for why WIPI2 puncta numbers are reduced and further argues against showing any C93E data.Is there a good reason to not show ATG16 puncta formation? This seems like the most direct test of the authors predictions, that these mutants will disrupt recruitment of the ATG16L1 complex?

C93E data have been removed, and see response to essential revisions.

Do the mutants in the ATG16L1 interaction site impact WIPI-membrane interaction? The authors interpret the reduction of some WIPI2 puncta in cells as evidence for a disruption in membrane binding and further, they suspect this is due to a cooperative interaction of WIPI with ATG16L1 and with membranes. To test this, they look at WIPI recruitment in their in vitro GUV assay in the presence of ATG16L1. It seems essential to do the same experiment without ATG16L1 though, to rule out the alternative interpretation that these mutations are having effects outside of ATG16L1 binding and perhaps are directly modifying the affinity of WIPI2 for phosphoinositide-rich membranes.

See response to essential revisions above.

Figure 3 – D is not indicated in legend. The text describes the outcomes of this experiment for individual mutations as they bind with similar or low "affinity", but the design of the experiment does not avail itself to this metric. Furthermore, no replicates are shown or quantification provided.

We thank the reviewer for catching the error. The text has been corrected to describe the experimental findings accurately. The figure legend has been amended to indicate experiments were performed in triplicates. The gel shown is representative of the other experiments. The authors thought the GUV assays showed more precise quantification and opted to use this to more precisely compare binding.

Figure 4 and Figure 6, axes labeled "Indensity".

This has been corrected. Thank you.

Figure 3 supplement (uncropped source data not included in main manuscript).

This figure has been added to the main manuscript.